# *Neopestalotiopsis* Species Associated with Flower Diseases of *Macadamia integrifolia* in Australia

**DOI:** 10.3390/jof7090771

**Published:** 2021-09-17

**Authors:** Kandeeparoopan Prasannath, Roger G. Shivas, Victor J. Galea, Olufemi A. Akinsanmi

**Affiliations:** 1Queensland Alliance for Agriculture & Food Innovation, The University of Queensland, Ecosciences Precinct, Dutton Park, QLD 4102, Australia; 2Centre for Crop Health, University of Southern Queensland, Toowoomba, QLD 4350, Australia; roger.shivas@daf.qld.gov.au; 3School of Agriculture & Food Sciences, The University of Queensland, Gatton, QLD 4343, Australia; v.galea@uq.edu.au

**Keywords:** amphisphaeriales, conidial morphology, flower blights, pestalotioid fungi, taxonomy, tree nut

## Abstract

Macadamia (*Macadamia integrifolia*) is native to eastern Australia and produces an edible nut that is extensively cultivated in commercial orchards in several countries. Little is known about the diversity of fungi associated with diseases of macadamia inflorescences. A survey of fungi associated with the dry flower disease of macadamia detected several isolates of *Neopestalotiopsis* (Pestalotiopsidaceae, Sordariomycetes). Five new species of *Neopestalotiopsis* were identified based on molecular phylogenetic analyses of concatenated gene sequences of the internal transcribed spacer (ITS), β-tubulin (TUB), and the translation elongation factor 1-alpha (TEF1α). The new species are named *Neopestalotiopsis drenthii*, *N. maddoxii*, *N. olumideae*, *N. vheenae,* and *N. zakeelii*, and are described by molecular, morphological, and cultural characteristics. The ecology of the isolates and their pathogenic, saprophytic, or commensal ability were not determined.

## 1. Introduction

Macadamia (*Macadamia integrifolia*) is a tree nut crop that is cultivated for its high-value kernel in tropical and subtropical regions in Australia, Asia, Africa, South America, and the U.S.A. Four species, *M. integrifolia*, *M. tetraphylla*, *M. ternifolia,* and *M. jansenii*, are native to Australia [1]. *Macadamia integrifolia* and *M. tetraphylla* produce edible kernels, whereas *M. ternifolia* and *M. jansenii* produce small and inedible nuts that contain high levels of cyanogenic glycosides [2]. Several new diseases caused by fungal and oomyceteous pathogens have been reported on macadamia with the expansion of its production area [3,4,5].

Diseases of flowers and fruit result in significant yield losses and poor-quality kernels [6,7,8]. A mature macadamia tree may produce over 10,000 racemes (inflorescences) at peak anthesis, with each raceme typically having 100–300 flowers (Figure 1a) [9,10]. Flower diseases can result in poor pollination efficiency, with less than 1% of the flowers producing fruit, as well as a reduction in the potential of the flower to bear fruit. Macadamia racemes may be affected by fungal pathogens at different developmental stages. Macadamia inflorescences have four growth stages: small green buds on the rachis (stage 1); florets that turn from light green to white and are partially up to fully open, with stamens that pull away from the stigmas (stage 2); fully opened flowers with sepals that turn brown at peak anthesis (stage 3); sepals that drop and flowers with swollen fertilized embryos (stage 4) [11]. Most of the diseases that affect macadamia inflorescences are flower blights [12]. A diversity of fungi has been associated with flower blights of macadamia in Australia [13], including species of *Botrytis* [14], *Cladosporium* [15], *Neopestalotiopsis,* and *Pestalotiopsis* [8].

The incidence of macadamia dry flower disease caused by *Pestalotiopsis macadamiae* and *Neopestalotiopsis macadamiae* is on the increase in Australian macadamia plantations [8]. Dry flower disease is characterized by the necrotic blight of flowers (Figure 1b). Akinsanmi et al. [16] suggested that multiple species of *Pestalotiopsis* and *Neopestalotiopsis* were responsible for dry flower epidemics in Australia.

*Pestalotiopsis* (Pestalotiopsidaceae, Sordariomycetes) was reclassified in 2014 and two new genera, *Neopestalotiopsis* and *Pseudopestalotiopsis*, were introduced [17]. Molecular phylogenies based on the combined sequences of three gene regions, including the internal transcribed spacer (ITS) region of rDNA, β-tubulin (TUB), and the translation elongation factor 1-alpha (TEF1α), have been used to delimit species within the pestalotioid genera (*Pestalotiopsis*, *Neopestalotiopsis,* and *Pseudo**pestalotiopsis*) [17]. Morphological identification of pestalotioid species is unreliable as species often have overlapping conidial measurements [18,19].

Several species of *Neopestalotiopsis* are phytopathogens in tropical and subtropical regions, causing leaf spot, dry flower, fruit rot, fruit scab, and trunk diseases on a range of crops [5,8,20,21,22,23,24,25,26]. Flower and leaf diseases on macadamia caused by *Neopestalotiopsis* spp. have been reported in Australia [5,8], Brazil [26], and China [25]. Many new pestalotioid species have been introduced in recent years [27,28,29,30,31,32,33]. Many unidentified isolates of *Neopestalotiopsis* were obtained from macadamia racemes with dry flower symptoms. However, there is little information about their identity and the diversity of fungi that cause dry flower disease on macadamia in Australia. The aim of this study was to identify the species of *Neopestalotiopsis* associated with the dry flowers of macadamia in Australia.

## 2. Materials and Methods

### 2.1. Sample Collection and Isolation

The isolates included in this study were collected from macadamia racemes with symptoms of dry flower disease. Samples were obtained from commercial macadamia orchards in Queensland (QLD) and New South Wales (NSW), Australia in 2019 and 2020. The samples were surface sterilized and incubated, as described by Akinsanmi et al. [8]. Monoconidial cultures of 13 isolates were established, as described by Akinsanmi et al. [34], and cryopreserved in a sterile 15% glycerol solution at −80 °C. Living cultures of the isolates were deposited in the Queensland Plant Pathology Herbarium (BRIP), Brisbane, Australia.

### 2.2. Cultural and Morphological Studies

Colony characteristics of cultures on ½-potato dextrose agar (PDA; Difco Laboratories, Franklin Lakes, NJ, USA.) medium were recorded after 7 d incubation at 25 °C. Fungal morphology was recorded from colonies grown in the dark for 14 d at 25 °C on PDA as well as on autoclaved pine needles on water agar. Fungal structures were examined in lactic acid on slide mounts under a Leica DM5500B compound microscope (Wetzlar, Germany) with Nomarski differential interference contrast illumination, and images were taken with a Leica DFC 500 camera. Measurements of at least 30 conidia and other fungal structures were taken at 1000× magnification. Novel species were registered in MycoBank [35].

### 2.3. DNA Extraction, PCR Amplification, and Sequencing

Genomic DNA was extracted from approx. 40 mg mycelium from colonies grown on PDA for 14 d. The mycelium was homogenized using TissueLyser (Qiagen, Chadstone, Australia) for 2 min at 30 Hz, and DNA was extracted using the BioSprint 96 DNA Plant Kit on a robotic platform (Qiagen, Chadstone, Australia). DNA concentration was determined with a BioDrop Duo spectrophotometer (BioDrop, Cambridge, England) and adjusted to 10 ng µL^−1^. The DNA of each isolate served as the template for the PCR amplifications using the reactions and thermal cyclic conditions described by Prasannath et al. [5]. Briefly, each reaction was performed in a 25 μL reaction volume, with 1 μL each of 10 μM forward and reverse primers, PCR reaction mix, and 2 μL of DNA template. PCR amplification was performed in SuperCycler Thermal Cycler (Kyratec, Wembley, Australia) at 95 °C for 2 min, followed by 35 cycles at 95 °C for 30 s, 55 °C for 30 s, and at 72 °C for 1 min, with a final extension step at 72 °C for 5 min. Three loci, ITS, TUB, and TEF1α, were amplified and sequenced using the primer pairs ITS4/ITS5 [36], BT2A/BT2B [37], and EF1-526F/EF1-1567R [38], respectively. The quality of PCR amplicons was checked on 1% agarose gel electrophoresis stained with GelRed (Biotium, Melbourne, Australia) under UV light by Molecular Imager GelDoc (Bio-Rad Laboratories Inc., Gladesville, Australia). The targeted PCR products were purified and sequenced in both directions at Macrogen Inc. (Seoul, South Korea).

### 2.4. Phylogenetic Analyses

The DNA sequences were assembled in Geneious Prime v. 2021.0.3 (Biomatters Ltd., San Diego, CA, USA.), manually trimmed and aligned to produce consensus sequences for each locus. The consensus sequences generated in this study were deposited in GenBank (Table 1). The sequences were compared against the NCBI GenBank nucleotide database using BLASTn to check the closest phylogenetic matches. The sequences of the ex-type isolates of the *Neopestalotiopsis* species were retrieved from GenBank (Table 1) and aligned with the sequences generated from our isolates using MAFFT v. 7.3.8.8 [39] in Geneious. Ambiguously aligned positions in each multiple alignment were excluded using Gblocks v. 0.91b [40]. The concatenated three-locus sequence dataset (ITS + TEF1α + TUB) of *Neopestalotiopsis* consisted of 63 taxa, with the outgroup taxon *Pestalotiopsis diversiseta* MFLUCC 12-0287 (Table 1). The combined sequence data matrix was manually improved with BioEdit v. 7.2.5 [41] and gaps were treated as missing data. Phylogenetic trees were generated from Maximum Likelihood (ML), Bayesian Inference (BI), and Maximum Parsimony (MP) analyses.

ML analysis was implemented using RAxML v. 8.2.11 [50] in Geneious. The search option was set to rapid bootstrapping, and the analysis was run using the GTR-GAMMAI evolution model with 1000 bootstrap iterations. BI analysis was conducted with MrBayes v. 3.2.1 [51] in Geneious to calculate posterior probabilities by the Markov Chain Monte Carlo (MCMC) method. The GTR-GAMMAI nucleotide substitution model was applied in BI analysis. Four MCMC chains were run simultaneously, starting from random trees for 1,000,000 generations. The temperature of the heated chain was set to 0.15 and trees were sampled every 200 generations until the average standard deviation of split frequencies reached 0.01 (stop value). Burn-in was set at 25%, after which the likelihood values were stationary. MP analysis was performed with PAUP v. 4.0b10 [52]. Trees were inferred using a heuristic search strategy, with 100 random stepwise addition and tree-bisection-reconnection (TBR) branch swapping. Max-trees were set to 5000 and bootstrap support values were evaluated for tree branches with 1000 replications [53]. Phylograms were visualized in FigTree v. 1.4.4 [54] and annotated in Adobe Illustrator 2021.

The Genealogical Concordance Phylogenetic Species Recognition (GCPSR) concept and a pairwise homoplasy index (PHI) test were used to determine species boundaries [55]. The PHI test was performed using SplitsTree4 v. 4.17.1 [56] to determine the recombination level within phylogenetically closely related species. The concatenated three-locus dataset (ITS + TEF1α + TUB) was used for the analyses. PHI test results (Fw) >0.05 indicated no significant recombination within the dataset. The relationships between closely related taxa were visualized in split graphs with both the Log-Det transformation and splits decomposition options.

## 3. Results

### 3.1. Phylogenetic Analyses

The concatenated sequence data matrix comprised 1367 base pairs (bp) (476 for ITS, 464 for TEF1α, and 427 for TUB), of which 935 bp were constant, 238 bp were parsimony-uninformative, and 174 bp were parsimony-informative. ML analysis yielded a best scoring tree, with a final ML optimization value of −6493.745 and the following model parameters: alpha—0.597, Π(A)—0.231, Π(C)—0.276, Π(G)—0.218, and Π(T)—0.274. Similar tree topologies were obtained by ML, BI, and MP methods, and the best scoring ML tree is shown in Figure 2. ML bootstrap values, BI posterior probabilities, and MP bootstrap values (MLBS/BIPP/MPBS) are given at nodes of the phylogram (Figure 2). The phylogenetic tree inferred from the concatenated alignment resolved the 13 *Neopestalotiopsis* isolates from symptomatic macadamia inflorescences (dry flower disease) into five well-supported monophyletic clades that represent novel species of *Neopestalotiopsis* (Figure 2).

### 3.2. Taxonomy

***Neopestalotiopsis drenthii*** Prasannath, Akinsanmi & R.G. Shivas, sp. nov. (Figure 3).

MycoBank: MB840916.

**Etymology**: Named after Andre Drenth, in recognition of his many contributions to the study of tropical and subtropical plant diseases.

**Type**: AUSTRALIA, Queensland, Mackay, from flower blight of *M. integrifolia*, 3 October 2019, *O.A. Akinsanmi* (**Holotype** BRIP 72264a, includes ex-type culture). GenBank: MZ303787 (ITS); MZ312680 (TUB); MZ344172 (TEF1α).

**Description**: *Conidiomata* pycnidial on PDA, globose, 200–400 µm diam., solitary or aggregated in clusters, exudes black conidial masses. *Conidiophores* reduced to conidiogenous cells. *Conidiogenous cells* ampulliform, hyaline, smooth, 5–20 × 2–5 μm. *Conidia* fusiform to ellipsoidal, straight or curved, 24–30 × 7–9 µm, 4-septate; basal cell conical, 4–6.5 µm, hyaline, smooth, thin-walled; with a single appendage filiform, unbranched, centric, 4–7 µm long; three median cells doliiform, 16–19 µm, smooth, versicolored, septa darker than the rest of the cell (second cell from base pale brown, 3.5–6.5 μm long; third cell medium to dark brown, 3.5–6.5 μm long; fourth cell medium to dark brown, 4–6 μm long); apical cell conical to subcylindrical, 3–5 µm long, hyaline, smooth, thin-walled; with 2–3 apical tubular appendages unbranched, filiform, 15–22 µm long. *Sexual morph* not seen.

**Culture characteristics**: Colonies on PDA after 7 d at 25 °C reach 80 mm diam., producing white aerial mycelia with copious pycnidia after two weeks; reverse cream.

**Habitat and distribution**: Racemes of *M. integrifolia* (Proteaceae); Australia.

**Other material examined**: AUSTRALIA, Queensland, Mackay, from flower blight of *M. integrifolia*, 3 October 2019, *O.A. Akinsanmi* (living culture, BRIP 72263a).

**Notes**: *Neopestalotiopsis drenthii* is closely related to *N. surinamensis*. A pairwise nucleotide comparison between *N. drenthii* ex-type strain (BRIP 72264a) and *N. surinamensis* ex-type strain (CBS 450.74) showed 2 bp differences (Identities 534/536, 2 gaps) in ITS, 0 bp differences (Identities 428/428, no gaps) in TUB, and 7 bp differences (Identities 474/481, no gaps) in TEF1α sequences in GenBank. *Neopestalotiopsis drenthii* and *N. surinamensis* have similar sized conidia, but tubular apical appendages of *N. drenthii* are shorter than the 18–27 µm of *N. surinamensis* [17].

***Neopestalotiopsis maddoxii*** Prasannath, Akinsanmi & R.G. Shivas, sp. nov. (Figure 4).

MycoBank: MB840917.

**Etymology**: Named after Craig Maddox, in recognition of his research contributions to macadamia crop protection in Australia.

**Type**: AUSTRALIA, Queensland, Alloway, from flower blight of *M. integrifolia*, 22 Sep. 2019, *Tim O’ Dale* (**Holotype** BRIP 72266a, includes ex-type culture). GenBank: MZ303782 (ITS); MZ312675 (TUB); MZ344167 (TEF1α).

**Description**: *Conidiomata* pycnidial on PDA, globose, 200–500 µm diam., solitary or aggregated in clusters, exudes dark slimy conidial droplets. *Conidiophores* reduced to conidiogenous cells. *Conidiogenous cells* ampulliform, hyaline, smooth, 5–15 × 2–5 μm. *Conidia* fusiform to clavate, straight or curved, 25–30 × 7–11 µm, 4-septate; basal cell conical, 3–5.5 µm, hyaline, smooth, thin-walled; with a single appendage filiform, unbranched, centric, 4–7 µm long; three median cells doliiform, 18–23 µm, smooth, versicolored, septa darker than the rest of the cell (second cell from base pale brown, 4.5–7.5 μm long; third cell medium to dark brown, 4.5–7.5 μm long; fourth cell medium to dark brown, 5–7 μm long); apical cell subcylindrical, 3–5 µm long, hyaline, smooth, thin-walled; with 3 apical tubular appendages unbranched, filiform, 15–27 µm long. *Sexual morph* not seen.

**Culture characteristics**: Colonies on PDA after 7 d at 25 °C reach 70 mm diam., with cream aerial mycelium, forming abundant pycnidia near the center after two weeks; reverse cream to buff.

**Habitat and distribution**: Racemes of *M. integrifolia* (Proteaceae); Australia.

**Other material examined**: AUSTRALIA, Queensland, Nambour, from flower blight of *M. integrifolia*, 22 Aug. 2019, *K. Prasannath* (living culture, BRIP 72260a); Bundaberg, from flower blight of *Macadamia* sp., 25 August 2019, *O.A. Akinsanmi* (living culture, BRIP 72262a); Gympie, from flower blight of *M. integrifolia*, 12 September 2019, *M. Boote* (living culture, BRIP 72272a); Maleny, from flower blight of *M. integrifolia*, 20 September 2019, *O.A. Akinsanmi* (living cultures, BRIP 72275a and BRIP 72284a).

**Notes**: *Neopestalotiopsis maddoxii* was sister to a clade containing *N. javaensis, N. mesopotamica,* and *N. rosae*. *Neopestalotiopsis maddoxii* differed from *N. javaensis* in ITS (Identities 534/539, 3 gaps); TUB (Identities 429/431, no gaps); TEF1α (Identities 480/486, 4 gaps). *Neopestalotiopsis maddoxii* differed from *N. mesopotamica* in ITS (Identities 534/539, 3 gaps); TUB (Identities 410/414, no gaps); TEF1α (Identities 466/472, no gaps). *Neopestalotiopsis maddoxii* differed from *N.*
*rosae* in ITS (Identities 534/539, 3 gaps); TUB (Identities 430/432, no gaps); TEF1α (Identities 478/481, no gaps). *Neopestalotiopsis maddoxii* is morphologically indistinguishable from *N. javaensis*, *N. mesopotamica,* and *N. rosae* [17].

***Neopestalotiopsis olumideae*** Prasannath, Akinsanmi & R.G. Shivas, sp. nov. (Figure 5).

MycoBank: MB840918.

**Etymology**: Named after Olumide Jeff-Ego, in recognition of her research contributions to macadamia diseases in Australia.

**Type**: AUSTRALIA, Queensland, Maleny, from flower blight of *M. integrifolia*, 20 Sep. 2019, *O.A. Akinsanmi* (**Holotype** BRIP 72273a, includes ex-type culture). GenBank: MZ303790 (ITS); MZ312683 (TUB); MZ344175 (TEF1α).

**Description**: *Conidiomata* pycnidial on PDA, globose, 200–400 µm diam., mostly solitary. *Conidiophores* reduced to conidiogenous cells. *Conidiogenous cells* ampulliform to cylindrical, hyaline, smooth, 5–15 × 2–5 μm. *Conidia* fusiform to ellipsoidal, straight or curved, 27–31 × 7–9 µm, 4-septate; basal cell conical, 4.5–7 µm, hyaline, smooth, thin-walled; with a single appendage filiform, unbranched, centric, 3–5 µm long; three median cells doliiform, 18–20 µm, smooth, versicolored, septa darker than the rest of the cell (second cell from base pale brown, 4–7 μm long; third cell medium to dark brown, 4–7 μm long; fourth cell medium to dark brown, 4.5–6.6 μm long); apical cell conical to subcylindrical, 3.5–5.5 µm long, hyaline, smooth, thin-walled; with 2–3 apical tubular appendages unbranched, filiform, 8–17 µm long. *Sexual morph* not seen.

**Culture characteristics**: Colonies on PDA after 7 d at 25 °C reach 60 mm diam., with whitish cottony aerial mycelium; reverse cream to buff.

**Habitat and distribution**: Racemes of *M. integrifolia* (Proteaceae); Australia.

**Other material examined**: AUSTRALIA, Queensland, Maleny, from flower blight of *M. integrifolia*, 20 September 2019, *O.A. Akinsanmi* (living culture, BRIP 72283a).

**Notes**: *Neopestalotiopsis olumideae* was phylogenetically close to *N. protearum, N. acrostichi* and *N. pernambucana*. *Neopestalotiopsis olumideae* differed from *N. protearum* in ITS (Identities 517/521, no gaps); TUB (Identities 436/438, no gaps); TEF1α (Identities 469/475, 3 gaps). *Neopestalotiopsis olumideae* differed from *N. acrostichi* in ITS (Identities 506/506, no gaps); TUB (Identities 435/436, no gaps); TEF1α (Identities 464/471, 1 gap). *Neopestalotiopsis olumideae* is morphologically indistinguishable from *N. protearum, N. acrostichi,* and *N. pernambucana* [17,30].

***Neopestalotiopsis vheenae*** Prasannath, Akinsanmi & R.G. Shivas, sp. nov. (Figure 6).

MycoBank: MB840919.

**Etymology**: Named after Vheena Mohankumar, in recognition of her research studies into fungal diseases of macadamia crops in Australia.

**Type**: AUSTRALIA, New South Wales, Rosebank, from flower blight of *M. integrifolia*, 16 October 2019, *P. Fraser* (**Holotype** BRIP 72293a, includes ex-type culture). GenBank: MZ303792 (ITS); MZ312685 (TUB); MZ344177 (TEF1α).

**Description**: *Conidiomata* pycnidial on PDA, globose, 200–500 µm diam., solitary or aggregated in clusters, exudes black slimy conidial droplets. *Conidiophores* reduced to conidiogenous cells. *Conidiogenous cells* ampulliform to cylindrical, hyaline, smooth, 5–10 × 3–5 μm. *Conidia* fusiform to clavate, straight or curved, 22–26 × 8–11 µm, 4-septate; basal cell conical, 3–5 µm, hyaline, smooth, thin-walled; with a single appendage filiform, unbranched, centric, 4–6.5 µm long; three median cells doliiform, 17–20 µm, smooth, versicolored, septa darker than the rest of the cell (second cell from base pale brown, 4–7 μm long; third cell medium to dark brown, 4–7 μm long; fourth cell medium to dark brown, 4.5–6.6 μm long); apical cell conical to subcylindrical, 3–4.5 µm long, hyaline, smooth, thin-walled; mostly with 3 apical tubular appendages unbranched, filiform, 15–25 µm long. *Sexual morph* not seen.

**Culture characteristics**: Colonies on PDA after 7 d at 25 °C reach 80 mm diam., with white dense aerial mycelium, pycnidia abundant; reverse buff.

**Habitat and distribution**: Racemes of *M. integrifolia* (Proteaceae); Australia.

**Notes**: *Neopestalotiopsis vheenae* ex-type strain (BRIP 72293a) had identical ITS, TUB, and TEF1α sequences to isolate BRIP 70210, which was previously identified as *N. clavispora* [5]. *Neopestalotiopsis vheenae* causes yellow halo spot of macadamia in Australia [5]. In this phylogenetic analysis, *N. vheenae* was sister to *N. sichuanensis*. BLASTn searches in GenBank showed that *N. vheenae* ex-type (BRIP 72293a) differed from *N. sichuanensis* ex-type (CFCC 5338) by 1 bp (Identities 510/511, no gaps) in ITS, 1 bp (Identities 422/423, no gaps) in TUB, and 10 bp (Identities 482/492, 6 gaps) in TEF1α sequences. *Neopestalotiopsis vheenae* is morphologically indistinguishable from *N. sichuanensis* [33].

***Neopestalotiopsis zakeelii*** Prasannath, Akinsanmi & R.G. Shivas, sp. nov. (Figure 7).

MycoBank: MB840920.

**Etymology**: Named after Mohamed Cassim Mohamed Zakeel, in recognition of his research into the diagnosis of new and emerging diseases on macadamia in Australia.

**Type**: AUSTRALIA, Queensland, Landsborough, from flower blight of *M. integrifolia*, 20 September 2019, *A. Woodford* (**Holotype** BRIP 72282a, includes ex-type culture). GenBank: MZ303789 (ITS); MZ312682 (TUB); MZ344174 (TEF1α).

**Description**: *Conidiomata* pycnidial on PDA, globose, 100–250 µm diam., mostly solitary. *Conidiophores* reduced to conidiogenous cells. *Conidiogenous cells* ampulliform, hyaline, smooth, 5–20 × 2–5 μm. *Conidia* fusiform to ellipsoidal, straight or curved, 27–33 × 7–9 µm, 4-septate; basal cell conical, 4.5–7 µm, hyaline, smooth, thin-walled; with a single appendage filiform, unbranched, centric, 7–10 µm long; three median cells doliiform, 17–22 µm, smooth, versicolored, septa darker than the rest of the cell (second cell from base pale brown, 4.5–7.5 μm long; third cell medium to dark brown, 4.5–7.5 μm long; fourth cell medium to dark brown, 5–7 μm long); apical cell conical to subcylindrical, 3–5.5 µm long, hyaline, smooth, thin-walled; with 2–3 apical tubular appendages unbranched, filiform, 25–37 µm long. *Sexual morph* not seen.

**Culture characteristics**: Colonies on PDA after 7 d at 25 °C reach 55 mm diam., with white sparse aerial mycelium; reverse cream.

**Habitat and distribution**: Racemes of *M. integrifolia* (Proteaceae); Australia.

**Other material examined**: AUSTRALIA, Queensland, Nambour, from flower blight of *M. integrifolia*, 22 August 2019, *K. Prasannath* (living culture, BRIP 72271a).

**Notes**: *Neopestalotiopsis zakeelii* is most closely related to *N. vitis* and *N. australis*. *Neopestalotiopsis zakeelii* (ex-type: BRIP 72282a) differed from *N. vitis* (ex-type: MFLUCC 15-1265) by 4 bp (Identities 416/420, no gaps) in ITS; 0 bp (Identities 319/319, no gaps) in TUB; 4 bp (Identities 369/373, no gaps) in TEF1α. *Neopestalotiopsis zakeelii* differed from *N. australis* (ex-type: CBS 114159) by 4 bp (Identities 526/530, no gaps) in ITS; 3 bp (Identities 430/433, no gaps) in TUB; 6 bp (Identities 478/484, no gaps) in TEF1α. *Neopestalotiopsis australis*, *N. vitis,* and *N. zakeelii* are morphologically indistinguishable [17,22].

There was no evidence of significant genetic recombination (Fw > 0.05) between the novel species of *Neopestalotiopsis* and closely related species (Figure 8). The results confirmed that these taxa were significantly different from the existing species of *Neopestalotiopsis*.

## 4. Discussion

Five novel species, *Neopestalotiopsis drenthii*, *N. maddoxii*, *N. olumideae*, *N. vheenae,* and *N. zakeelii*, were discovered in isolates obtained from macadamia inflorescences with dry flower disease and subsequently described. There was no evidence of significant genetic recombination events between these species and their closest relatives.

Pestalotioid fungi (Pestalotiopsidaceae, Sordariomycetes) are species-rich asexual taxa, which are common pathogens on many crops [30,32,57]. Multi-locus phylogenetic analyses segregated *Neopestalotiopsis* and *Pseudopestalotiopsis* from *Pestalotiopsis* [17]. These three genera are morphologically similar in having 5-celled conidia with tubular apical appendages [17]. Norphanphoun et al. [30] found that concatenated gene sequences of ITS, TUB, and TEF1α resolved *Pestalotiopsis* and *Pesudopestalotiopsis*, while additional genes may be required to provide a better delimitation of *Neopestalotiopsis* spp.

There were 49 species names recognized in *Neopestalotiopsis* [58] prior to the five species described in this study. *Neopestalotiopsis drenthii*, *N. maddoxii*, *N. olumideae*, *N. vheenae,* and *N. zakeelii* formed well-supported monophyletic clades in the phylogenetic analysis. The topology of our phylogeny is similar to those generated in earlier studies [29,57].

*Pestalotiopsis macadamiae* and *N. macadamiae* were first reported as the causal fungi of dry flower disease of macadamia by Akinsanmi et al. [8]. These fungi are considered endemic to Australia and have likely co-evolved with macadamia. *Pestalotiopsis macadamiae* has been reported outside Australia on macadamia leaves in China [59]. The present study found a high diversity of *Neopestalotiopsis* spp. on macadamia racemes with dry flower symptoms. It is not known whether *Neopestalotiopsis drenthii*, *N. maddoxii*, *N. olumideae*, *N. vheenae,* and *N. zakeelii* are pathogens or saprobes. The role of these fungi in the pathogenicity of dry flower disease remains to be demonstrated by Koch’s postulates. Living cultures of *N. drenthii*, *N. maddoxii*, *N. olumideae*, *N. vheenae,* and *N. zakeelii* are accessible in the BRIP culture collection for future research and comparative studies that may lead to the elucidation of their lifecycle and their role in dry flower disease.

## 5. Conclusions

Five fungal species, *Neopestalotiopsis drenthii*, *N. maddoxii*, *N. olumideae*, *N. vheenae,* and *N. zakeelii,* were described and illustrated. These fungi were isolated from inflorescences of macadamia with dry flower disease in Australia. The role that *N. drenthii*, *N. maddoxii*, *N. olumideae*, *N. vheenae,* and *N. zakeelii* play in dry flower disease of macadamia has yet to be determined. Hence, the pathogenicity of these novel species on macadamia racemes should be examined. Living cultures of the fungi are available for further study.

## Figures and Tables

**Figure 1 jof-07-00771-f001:**
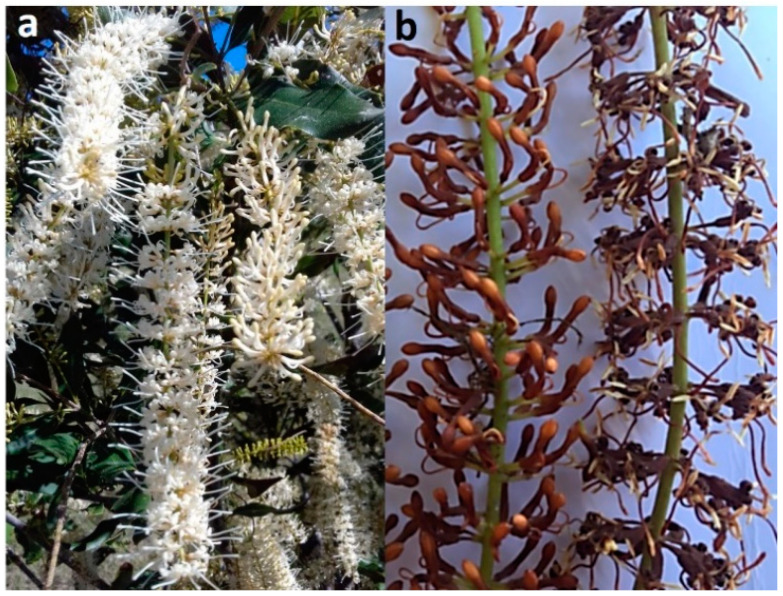
Macadamia inflorescences (racemes). (**a**) Pendant racemes in tree canopy, and (**b**) dry flower disease.

**Figure 2 jof-07-00771-f002:**
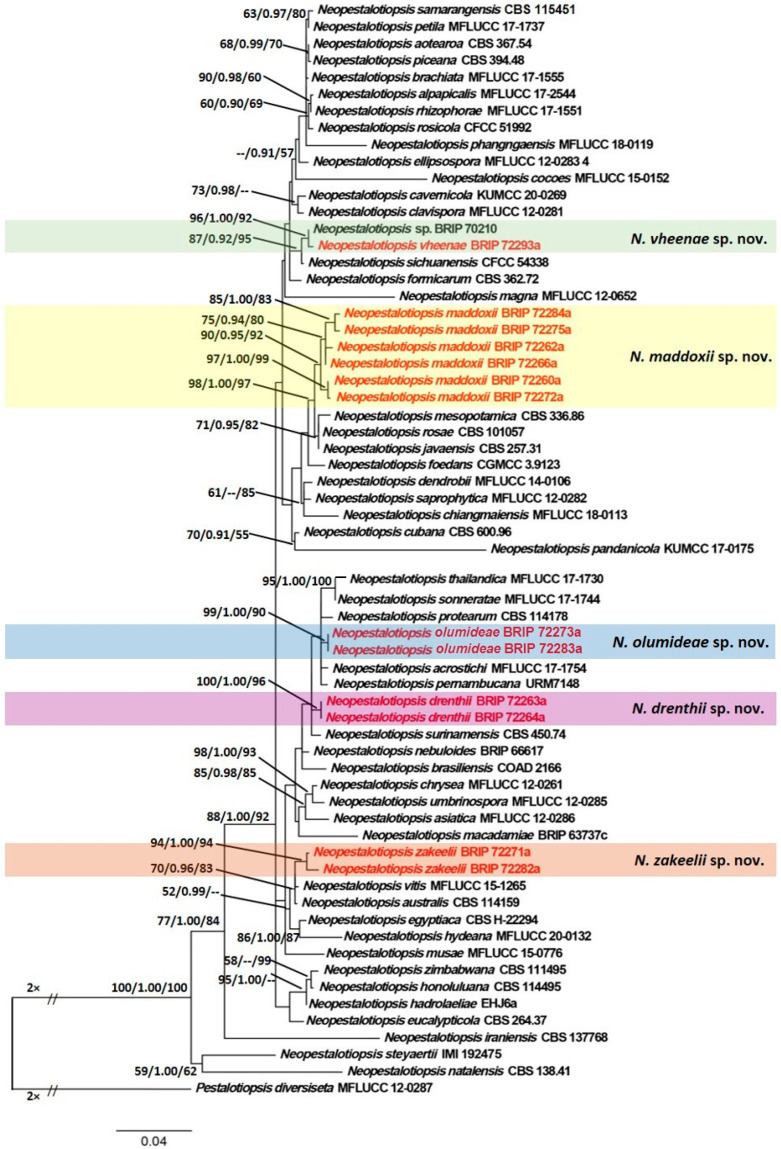
Maximum Likelihood tree topology of *Neopestalotiopsis* based on a combined multi-locus alignment (ITS + TEF1α + TUB). *Pestalotiopsis diversiseta* (MFLUCC 12-0287) was used as an outgroup taxon. Maximum Likelihood bootstrap support values (>50%), Bayesian Inference posterior probabilities (>90%), and Maximum Parsimony bootstrap proportions (>50%) are displayed at the nodes, respectively. Strains of the newly described species are depicted in red.

**Figure 3 jof-07-00771-f003:**
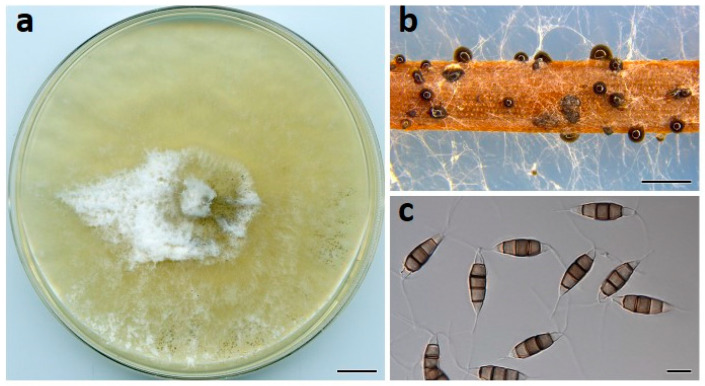
*Neopestalotiopsis drenthii* (BRIP 72264a). (**a**) Two-week-old colony on PDA, (**b**) conidiomata on pine needle agar, and (**c**) conidia. Scale bars: a = 1 cm; b = 1 mm; c = 10 µm.

**Figure 4 jof-07-00771-f004:**
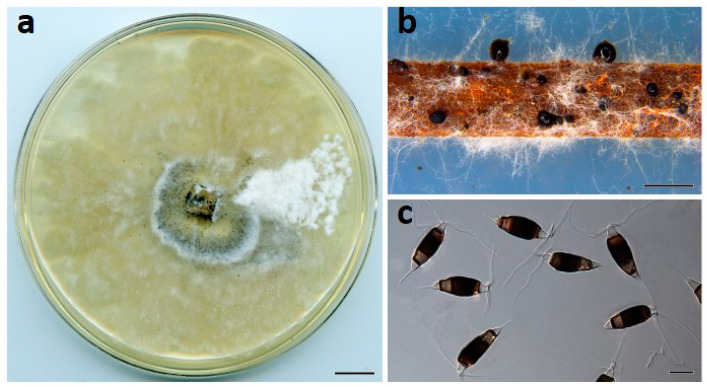
*Neopestalotiopsis maddoxii* (BRIP 72266a). (**a**) Two-week-old colony on PDA, (**b**) conidiomata on pine needle agar, and (**c**) conidia. Scale bars: a = 1 cm; b = 1 mm; c = 10 µm.

**Figure 5 jof-07-00771-f005:**
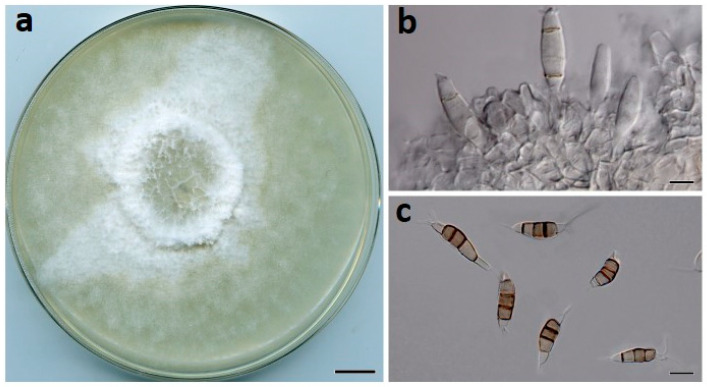
*Neopestalotiopsis olumideae* (BRIP 72273a). (**a**) Two-week-old colony on PDA, (**b**) conidiogenous cells, and (**c**) conidia. Scale bars: a = 1 cm; b–c = 10 µm.

**Figure 6 jof-07-00771-f006:**
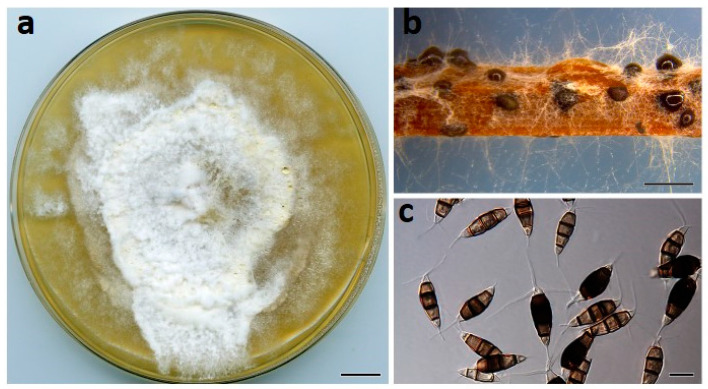
*Neopestalotiopsis vheenae* (BRIP 72293a). (**a**) Two-week-old colony on PDA, (**b**) conidiomata on pine needle agar, and (**c**) conidia. Scale bars: a = 1 cm; b = 1 mm; c = 10 µm.

**Figure 7 jof-07-00771-f007:**
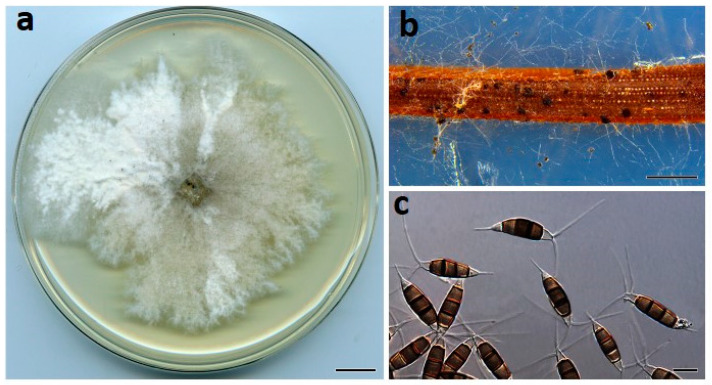
*Neopestalotiopsis zakeelii* (BRIP 72282a). (**a**) Two-week-old colony on PDA, (**b**) conidiomata on pine needle agar, and (**c**) conidia. Scale bars: a = 1 cm; b = 1 mm; c = 10 µm.

**Figure 8 jof-07-00771-f008:**
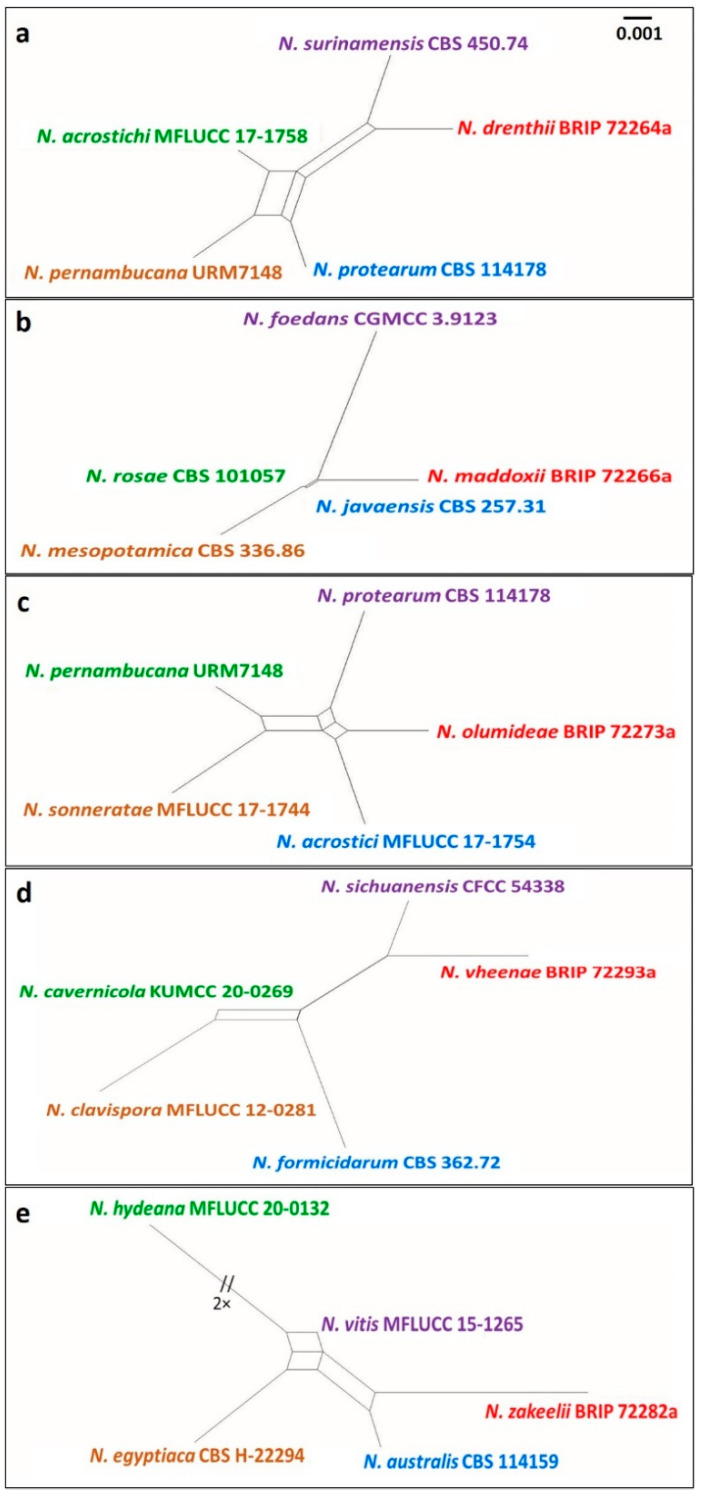
Split graphs showing the result of PHI test of (**a**) *Neopestalotiopsis drenthii* (Fw = 0.735), (**b**) *N. maddoxii* (Fw = 0.118), (**c**) *N. olumideae* (Fw = 0.581), (**d**) *N. vheenae* (Fw = 0.876), and (**e**) *N. zakeelii* (Fw = 0.412) with their most closely related species using Log-Det transformation and splits decomposition options. The new taxon in each graph is shown in red font.

**Table 1 jof-07-00771-t001:** *Neopestalotiopsis* species and isolates used in the phylogenetic analyses, with GenBank accession numbers.

Species	Strain ^1^	Host/Substrate	Location	GenBank Accession Numbers ^2^			Reference
				**ITS**	**TUB**	**TEF1α**	
*Neopestalotiopsis acrostichi*	MFLUCC 17–1754 ^T^	*Acrostichum aureum*	Thailand	MK764272	MK764338	MK764316	[30]
*N. alpapicalis*	MFLUCC 17–2544 ^T^	*Rhizophora mucronata*	Thailand	MK357772	MK463545	MK463547	[28]
*N. aotearoa*	CBS 367.54 ^T^	Canvas	New Zealand	KM199369	KM199454	KM199526	[17]
*N. asiatica*	MFLUCC 12–0286 ^T^	*Prunus dulcis*	China	JX398983	JX399018	JX399049	[17]
*N. australis*	CBS 114159 ^T^	*Telopea* sp.	Australia	KM199348	KM199432	KM199537	[17]
*N. brachiata*	MFLUCC 17–555 ^T^	*Rhizophora apiculata*	Thailand	MK764274	MK764340	MK764318	[30]
*N. brasiliensis*	COAD 2166 ^T^	*Psidium guajava*	Brazil	MG686469	MG692400	MG692402	[42]
*N. cavernicola*	KUMCC 20–0269 ^T^	Cave	China	MW545802	MW557596	MW550735	[29]
*N. chiangmaiensis*	MFLUCC 18–0113 ^T^	* Pandanus * sp.	Thailand	N/A	MH412725	MH388404	[43]
*N. chrysea*	MFLUCC 12–0261 ^T^	Dead leaves	China	JX398985	JX399020	JX399051	[19]
*N. clavispora*	MFLUCC 12–0281 ^T^	* Magnolia * sp.	China	JX398979	JX399014	JX399045	[19]
*N. cocoes*	MFLUCC 15–0152 ^T^	* Cocos nucifera *	Thailand	KX789687	N/A	KX789689	[44]
* N. cubana *	CBS 600.96 ^T^	Leaf Litter	Cuba	KM199347	KM199438	KM199521	[17]
* N. * *dendrobii*	MFLUCC 14–0106 ^T^	* Dendrobium cariniferum *	Thailand	MK993571	MK975835	MK975829	[45]
*N. drenthii*	BRIP 72263a	* Macadamia integrifolia *	Australia	MZ303786	MZ312679	MZ344171	This study
	BRIP 72264a ^T^	* Macadamia integrifolia *	Australia	MZ303787	MZ312680	MZ344172	This study
*N. egyptiaca*	CBS H–22294 ^T^	* Mangifera indica *	Egypt	KP943747	KP943746	KP943748	[46]
*N. ellipsospora*	MFLUCC 12–0283 ^T^	Dead plant material	China	JX398980	JX399016	JX399047	[19]
*N. eucalypticola*	CBS 264.37 ^T^	* Eucalyptus globulus *	N/A	KM199376	KM199431	KM199551	[17]
*N. foedans*	CGMCC 3.9123 ^T^	Mangrove plant	China	JX398987	JX399022	JX399053	[19]
*N. formicidarum*	CBS 362.72 ^T^	Dead Formicidae (ant)	Cuba	KM199358	KM199455	KM199517	[17]
*N. hadrolaeliae*	EHJ6a	*Cattleya jongheana*	Brazil	MK454709	MK465120	MK465122	[27]
*N. honoluluana*	CBS 114495 ^T^	* Telopea * sp.	USA	KM199364	KM199457	KM199548	[17]
*N. hydeana*	MFLUCC 20–0132 ^T^	*Artocarpus heterophyllus*	Thailand	MW266069	MW251119	MW251129	[32]
*N. iranensis*	CBS 137768 ^T^	*Fragaria ananassa*	Iran	KM074048	KM074057	KM074051	[20]
*N. javaensis*	CBS 257.31 ^T^	*Cocos nucifera*	Java	KM199357	KM199437	KM199543	[17]
*N. macadamiae*	BRIP 63737c ^T^	*Macadamia* sp.	Australia	KX186604	KX186654	KX186627	[8]
*N. maddoxii*	BRIP 72260a	*Macadamia integrifolia*	Australia	MZ303780	MZ312673	MZ344165	This study
	BRIP 72262a	*Macadamia integrifolia*	Australia	MZ303781	MZ312674	MZ344166	This study
	BRIP 72266a ^T^	*Macadamia integrifolia*	Australia	MZ303782	MZ312675	MZ344167	This study
	BRIP 72272a	*Macadamia integrifolia*	Australia	MZ303783	MZ312676	MZ344168	This study
	BRIP 72275a	*Macadamia integrifolia*	Australia	MZ303784	MZ312677	MZ344169	This study
	BRIP 72284a	*Macadamia integrifolia*	Australia	MZ303785	MZ312678	MZ344170	This study
*N. magna*	MFLUCC 12–652 ^T^	* Pteridium * sp.	France	KF582795	KF582793	KF582791	[17]
*N. mesopotamica*	CBS 336.86 ^T^	* Pinus brutia *	Iraq	KM199362	KM199441	KM199555	[17]
*N. musae*	MFLUCC 15–0776 ^T^	* Musa * sp.	Thailand	KX789683	KX789686	KX789685	[44]
*N. natalensis*	CBS 138.41 ^T^	* Acacia mollissima *	South Africa	KM199377	KM199466	KM199552	[17]
*N. nebuloides*	BRIP 66617 ^T^	* Sporobolus elongatus *	Australia	MK966338	MK977632	MK977633	[31]
*N. olumideae*	BRIP 72273a ^T^	* Macadamia integrifolia *	Australia	MZ303790	MZ312683	MZ344175	This study
	BRIP 72283a	* Macadamia integrifolia *	Australia	MZ303791	MZ312684	MZ344176	This study
*N. pandanicola*	KUMCC 17–0175	* Pandanus * sp.	China	N/A	MH412720	MH388389	[43]
*N. pernambucana*	URM7148	* Vismia guianensis *	Brazil	KJ792466	N/A	KU306739	[47]
*N. petila*	MFLUCC 17–1737 ^T^	* Rhizophora mucronata *	Thailand	MK764275	MK764341	MK764319	[30]
*N. phangngaensis*	MFLUCC 18–0119 ^T^	* Pandanus * sp.	Thailand	MH388354	MH412721	MH388390	[43]
*N. piceana*	CBS 394.48 ^T^	* Picea * sp.	UK	KM199368	KM199453	KM199527	[17]
*N. protearum*	CBS 114178 ^T^	* Leucospermum cuneiforme *	Zimbabwe	JN712498	KM199463	LT853201	[17]
*N. rhizophorae*	MFLUCC 17–1551 ^T^	* Rhizophora mucronata *	Thailand	MK764278	MK764344	MK764322	[30]
*N. rosae*	CBS 101057 ^T^	* Rosa * sp.	New Zealand	KM199359	KM199429	KM199523	[17]
*N. rosicola*	CFCC 51992 ^T^	* Rosa chinensis *	China	KY885239	KY885245	KY885243	[48]
*N. samarangensis*	CBS 115451	Unidentified tree	China	KM199365	KM199447	KM199556	[49]
*N. saprophytica*	MFLUCC 12–0282 ^T^	* Magnolia * sp.	China	JX398982	JX399017	JX399048	[19]
*N. sichuanensis*	CFCC 54338 ^T^	*Castanea mollissima*	China	MW166231	MW218524	MW199750	[33]
*N. sonneratae*	MFLUCC 17–1744 ^T^	* Sonneronata alba *	Thailand	MK764279	MK764345	MK764323	[30]
*N. steyaertii*	IMI 192475 ^T^	* Eucalyptus viminalis *	Australia	KF582796	KF582794	KF582792	[17]
*N. surinamensis*	CBS 450.74 ^T^	Soil under	Suriname	KM199351	KM199465	KM199518	[17]
		* Elaeis guineensis *					
*N. thailandica*	MFLUCC 17–1730 ^T^	* Rhizophora mucronata *	Thailand	MK764281	MK764347	MK764325	[30]
*N. umbrinospora*	MFLUCC 12–0285 ^T^	Unidentified plant	China	JX398984	JX399019	JX399050	[19]
*N. vheenae*	BRIP 72293a ^T^	* Macadamia integrifolia *	Australia	MZ303792	MZ312685	MZ344177	This study
	BRIP 70210	* Macadamia integrifolia *	Australia	MN114212	MN114214	MN114213	[5] ^3^
*N. vitis*	MFLUCC 15–1265 ^T^	* Vitis vinifera *	China	KU140694	KU140685	KU140676	[22]
*N. zakeelii*	BRIP 72271a	* Macadamia integrifolia *	Australia	MZ303788	MZ312681	MZ344173	This study
	BRIP 72282a ^T^	* Macadamia integrifolia *	Australia	MZ303789	MZ312682	MZ344174	This study
*N. zimbabwana*	CBS 111495 ^T^	* Leucospermum *	Zimbabwe	JX556231	KM199456	KM199545	[17]
		* cunciforme *					
*Pestalotiopsis diversiseta*	MFLUCC 12–0287 ^T^	Dead plant material	China	JX399009	JX399040	JX399073	[19]

^1^ BRIP: Queensland Plant Pathology Herbarium, Australia; CBS: Culture collection of the Centraalbureau voor Schimmelcultures, Fungal Biodiversity Centre, Utrecht, The Netherlands; CFCC: China Forestry Culture Collection Center, Research Institute of Forest Ecology, Environment and Protection, Beijing, China; CGMCC: China General Microbiological Culture Collection Center, Institute of Microbiology, Chinese Academy of Sciences, Beijing, China; COAD: Culture collection Coleção Octávio Almeida Drummond of the Universidade Federal de Viçosa, Viçosa, Brazil; HGUP: Plant Pathology Herbarium of Guizhou University, Guizhou, China; IMI: Culture collection of CABI Europe UK Centre, Egham, UK; KUMCC: Culture Collection of Kunming Institute of Botany, Chinese Academy of Sciences, Kunming, China; MFLUCC: Mae Fah Luang University Culture Collection, Chiang Rai, Thailand; URM: The Father Camille Torrend Herbarium, Pernambuco, Brazil. ^2^ ITS: internal transcribed spacer; TUB: β-tubulin; TEF1α: translation elongation factor 1-α. ^3^ as *Neopestalotiopsis clavispora*. Ex-type strains are labeled with ^T^. N/A: Not available.

## Data Availability

All sequence data are available in NCBI GenBank following the accession numbers in the manuscript.

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
