# Peer review of "Neopestalotiopsis Species Associated with Flower Diseases of Macadamia integrifolia in Australia"

_jof, 2021, doi:10.3390/jof7090771_

Round 1
Reviewer 1 Report
Here is the review of the paper entitled "Neopestalotiopsis Species Associated with Flower Diseases of Macadamia integrifolia in Australia" written by Kandeeparoopan Prasannath and co-authors.
The authors researched fungi associated with dry flower disease of Macadamia integrifolia. Several Neopestalotiopsis (Pestalotiopsidaceae, Sordariomycetes) isolates were found and identified based on morphological, molecular and culture characteristcs. All known sequences of type specimens of Neopestalotiopsis were downloaded from Genbank for the multigene phylogenetic analysis of internal transcribed spacer (ITS), β-tubulin (TUB) and translation elongation factor 1-alpha (TEF1α) sequences. Newly obtained sequences of Australian isolates of Neopestalotiopsis were added to the dataset, phylogenetically analysed and five new species were described. All newly described species formed well-supported monophyletic clades in the phylogenetic analysis.
The morphological and molecular research methods used in the study are suitable, well-produced, and described. Authors fully followed the newest International code of nomenclature for algae, fungi, and plants. The topic is interesting for JoF audience. The English language used in the manuscript is fine.
Suggested corrections/additions are included in the attached review of the manuscript file.
I find the manuscript suitable for publication after minor revision!
Best, Reviewer

Author Response
Point 1: Please describe briefly PCR procedure here to avoid need to look at Prasannath et al.
Response 1: As suggested, the PCR procedure has been described (Lines 100-104).
Point 2: Why Macadamia sp.? Not Macadamia integrifolia?
Response 2: As suggested, ‘Macadamia sp.’ has been changed to ‘Macadamia integrifolia’
in Table 1 and the text (Lines 192, 206, 225, 241, 265, 279, 298, 313, 330, 344).
Point 3: Are there any other differentiating morphological characteristics between the two species? Please comment on this.
Response 3: A distinct morphological character between those two species has been added (Lines 214-215).
Point 4: Why Macadamia sp.? Not Macadamia integrifolia? check other Macadamia sp. in the text.
Response 4: ‘Macadamia sp.’ has been changed to ‘Macadamia integrifolia’ in the text (Lines 192, 206, 225, 241, 265, 279, 298, 313, 330, 344) and Table 1.
Point 5: macademia -> macadamia
Response 5: The spelling mistake has been corrected (Line 398).

Reviewer 2 Report
Well written paper. Methods are sound. Need to include statement in the abstract that the ecology of the isolates, pathogenic, saprophytic, or commensal, was not determined. Also need to clarify in introduction the reason for the work. It is well explained in the conclusion, but needs another sentence to differentiate between what had been previously determined and what still needed to be done.
Author Response
Point 1: Need to include statement in the abstract that the ecology of the isolates, pathogenic, saprophytic, or commensal, was not determined.
Response 1: As suggested, the statement has been included in the Abstract (Lines 20-21).
Point 2: Also need to clarify in introduction the reason for the work.
Response 2: The reason for the work has been given in the Introduction (Lines 68-71).
Point 3: It is well explained in the conclusion, but needs another sentence to differentiate between what had been previously determined and what still needed to be done.
Response 3: This information was provided in the discussion (Lines 388-393). Additional sentence has been added in the Conclusion (Lines 399-400).
